# Leveraging Optimization for Adaptive Attacks on Image Watermarks

**Nils Lukas, Abdulrahman Diaa\*, Lucas Fenaux\*, Florian Kerschbaum**
University of Waterloo, Canada
`{nlukas,abdulrahman.diaa,lucas.fenaux,`
`florian.kerschbaum}@uwaterloo.ca`

## Abstract

Untrustworthy users can misuse image generators to synthesize high-quality deep-fakes and engage in unethical activities. Watermarking deters misuse by marking generated content with a hidden message, enabling its detection using a secret watermarking key. A core security property of watermarking is robustness, which states that an attacker can only evade detection by substantially degrading image quality. Assessing robustness requires designing an adaptive attack for the specific watermarking algorithm. When evaluating watermarking algorithms and their (adaptive) attacks, it is challenging to determine whether an adaptive attack is optimal, i.e., the best possible attack. We solve this problem by defining an objective function and then approach adaptive attacks as an optimization problem. The core idea of our adaptive attacks is to replicate secret watermarking keys locally by creating *surrogate keys* that are differentiable and can be used to optimize the attack's parameters. We demonstrate for Stable Diffusion models that such an attacker can break all five surveyed watermarking methods at no visible degradation in image quality. Optimizing our attacks is efficient and requires less than 1 GPU hour to reduce the detection accuracy to 6.3% or less. Our findings emphasize the need for more rigorous robustness testing against adaptive, learnable attackers.

## 1 Introduction

Deepfakes are images synthesized using deep image generators that can be difficult to distinguish from real images. While deepfakes can serve many beneficial purposes if used ethically, for example, in medical imaging (Akrout et al., 2023) or education (Peres et al., 2023), they also have the potential to be *misused* and erode trust in digital media. Deepfakes have already been used in disinformation campaigns (Boneh et al., 2019; Barrett et al., 2023) and social engineering attacks (Mirsky & Lee, 2021), highlighting the need for methods that control the misuse of deep image generators.

Watermarking offers a solution to controlling misuse by embedding hidden messages into all generated images that are later detectable using a secret watermarking key. Images detected as deepfakes can be flagged by social media platforms or news agencies, which can mitigate potential harm (Grinbaum & Adomaitis, 2022). Providers of large image generators such as Google have announced the deployment of their own watermarking methods (Gowal & Kohli, 2023) to enable the detection of deepfakes and promote the ethical use of their models, which was also declared as one of the main goals in the US government's "AI Executive Order" (Federal Register, 2023).

A core security property of watermarking is *robustness*, which states that an attacker can evade detection only by substantially degrading the image's quality. While several watermarking methods have been proposed for image generators (Wen et al., 2023; Zhao et al., 2023; Fernandez et al., 2023), none of them are certifiably robust (Bansal et al., 2022) and instead, robustness is tested empirically using a limited set of known attacks. Claimed security properties of previous watermarking methods have been broken by novel attacks (Lukas et al., 2022), and no comprehensive method exists to validate robustness, which causes difficulty in trusting the deployment of watermarking in practice. We propose testing the robustness of watermarking by defining robustness using objective

---

*Equal Contribution

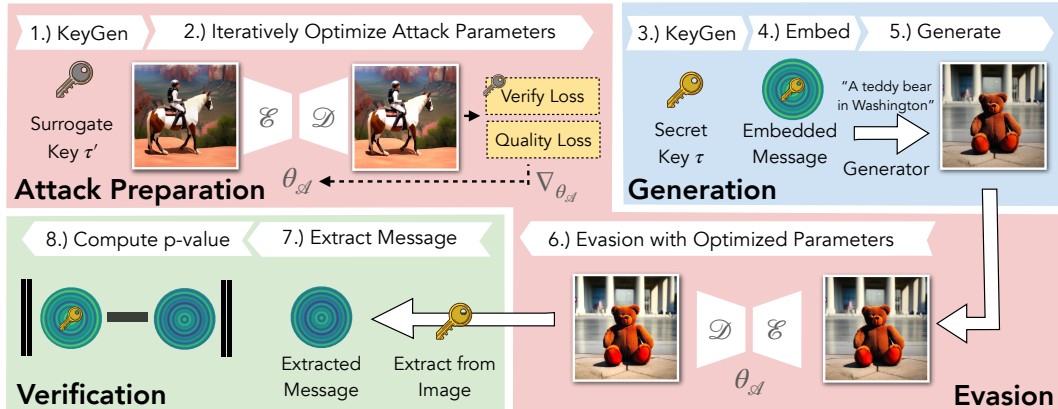

Figure 1: An overview of our adaptive attack pipeline. The attacker prepares their attack by generating a surrogate key and leveraging optimization to find optimal attack parameters $\theta_\mathcal{A}$ (illustrated here as an encoder $\mathcal{E}$ and decoder $\mathcal{D}$) for any message. Then, the attacker generates watermarked images and applies a modification using their optimized attack to evade detection. The attack is successful if the verification procedure cannot detect the watermark in high-quality images.

function and approaching adaptive attacks as an optimization problem. Adaptive attacks are specific to the watermarking algorithm used by the defender but have no access to the secret watermarking key. Knowledge of the watermarking algorithm enables the attacker to consider a range of *surrogate* keys similar to the defender's key. This also presents a challenge for optimization since the attacker only has imperfect information about the optimization problem. Adaptive attackers had previously been shown to break the robustness of watermarking for image classifiers (Lukas et al., 2022), but attacks had to be handcrafted against each watermarking method. Finding attack parameters through an optimization process can be challenging when the watermarking method is not easily optimizable, for instance, when it is not differentiable. Our attacks leverage optimization by approximating watermark verification through a differentiable process. Figure 1 shows that our adaptive attacker can prepare their attacks before the provider deploys their watermark. We show that adaptive, *learnable* attackers, whose parameters can be optimized efficiently, can evade watermark detection for 1 billion parameter Stable Diffusion models at a negligible degradation in image quality.

## 2 BACKGROUND

**Latent Diffusion Models** (LDMs) are state-of-the-art generative models for image synthesis (Rombach et al., 2022). Compared to Diffusion Models (Sohl-Dickstein et al., 2015), LDMs operate in a latent space using fixed, pre-trained autoencoder consisting of an image encoder $\mathcal{E}$ and a decoder $\mathcal{D}$. LDMs use a forward and reverse diffusion process across $T$ steps. In the forward pass, real data point $x_0$ is encoded into a latent point $z_0 = \mathcal{E}(x_0)$ and is progressively corrupted into noise via Gaussian perturbations. Specifically,

$$q(z_t|z_{t-1}) = \mathcal{N}\left(z_t; \sqrt{1-\beta_t}z_{t-1}, \beta_t\mathbf{I}\right), \quad t \in \{0, 1, \ldots, T-1\}, \tag{1}$$

where $\beta_t$ is the scheduled variance. In the reverse process, a neural network $f_\theta$ guides the denoising, taking $z_t$ and time-step $t$ as inputs to predict $z_{t-1}$ as $f_\theta(x_t, t)$. The model is trained to minimize the mean squared error between the predicted and actual $z_{t-1}$. The outcome is a latent $\hat{z}_0$ resembling $z_0$ that can be decoded into $\hat{x}_0 = \mathcal{D}(z_0)$. Synthesis in LDMs can be conditioned with textual prompts.

### 2.1 WATERMARKING

Watermarking embeds a hidden signal into a medium, such as images, using a secret watermarking key that is later extractable using the same secret key. Watermarking can be characterized by the medium used by the defender to verify the presence of the hidden signal. White-box and black-

box watermarking methods assume access to the model's parameters or query access via an API, respectively, and have been used primarily for Intellectual Property protection (Uchida et al., 2017)[1].

*No-box* watermarking (Lukas & Kerschbaum, 2023) assumes a more restrictive setting where the defender only knows the generated content but does not know the query used to generate the image. This type of watermarking has been used to control misuse by having the ability to detect any image generated by the provided image generator (Gowal & Kohli, 2023). Given a generator's parameters $\theta_G$, a no-box watermarking method defines the following three procedures.

- $\tau \leftarrow \text{KEYGEN}(\theta_G)$: A randomized function to generate a watermarking key $\tau$.

- $\theta_G^* \leftarrow \text{EMBED}(\theta_G, \tau, m)$: For a generator $\theta_G$, a watermarking key $\tau$ and a message $m$, return parameters $\theta_G^*$ of a *watermarked* generator[2] that only generates watermarked images.

- $p \leftarrow \text{VERIFY}(x, \tau, m)$: This function (i) extracts a message $m'$ from $x$ using $\tau$ and (ii) returns the $p$-value to reject the null hypothesis that $m$ and $m'$ match by random chance.

A watermarking method is a set of algorithms that specify (KEYGEN, EMBED, VERIFY). A watermark is a hidden signal in an image that can be mapped to a message $m$ using a secret key $\tau$. The key refers to secret random bits of information used in the randomized verification algorithm to detect a message. Adaptive attackers know the watermarking method but not the key message pair. Carlini & Wagner (2017) first studied adaptive attacks in the context of adversarial attacks.

In this paper, we denote the similarity between two messages by their $L_1$-norm difference. We use more meaningful similarity measures when $\mathcal{M}$ allows it, such as the Bit-Error-Rate (BER) when the messages consist of bits. A watermark is *retained* in an image if the verification procedure returns $p < 0.01$, following Wen et al. (2023). Adi et al. (2018) specify the requirements for trustworthy watermarking, and we focus on two properties: Effectiveness and robustness. Effectiveness states that a watermarked generator has a high image quality while retaining the watermark, and robustness means that a watermark is retained in an image unless the image's quality is substantially degraded. We refer to Lukas & Kerschbaum (2023) for security games encoding effectiveness and robustness.

## 2.2 Watermarking for Image Generators

Several works propose no-box watermarking methods to prevent misuse for two types of image generators: Generative Adversarial Networks (GANs) (Goodfellow et al., 2020) and Latent Diffusion Models (LDMs) (Rombach et al., 2022). We distinguish between *post-hoc* watermarking methods that apply an imperceptible modification to an image and *semantic* watermarks that modify the output distribution of an image generator and are truly "invisible" (Wen et al., 2023).

For post-hoc watermarking, traditional methods hide messages using the Discrete Wavelet Transform (DWT) and Discrete Wavelet Transform with Singular Value Decomposition (DWT-SVD) (Cox et al., 2007) and are currently used for Stable Diffusion. RivaGAN (Zhang et al., 2019) watermarks by training a deep neural network adversarially to stamp a pattern on an image. Yu et al. (2020; 2021) propose two methods that modify the generator's training procedure but require expensive re-training from scratch. Lukas & Kerschbaum (2023) propose a watermarking method for GANs that can be embedded into a pre-trained generator. Zhao et al. (2023) propose a general method to watermark diffusion models (WDM) that uses a method similar to Yu et al. (2020), which trains an autoencoder to stamp a watermark on all training data before also re-training the generator from scratch. Fernandez et al. (2023) pre-train an autoencoder to encode hidden messages into the training data and embed the watermark by fine-tuning the decoder $\mathcal{D}$ component of the LDM.

Wen et al. (2023) are the first to propose a semantic watermarking method for LDMs they call Tree-Rings Watermarks (TRW). The idea is to mark the initial noise $x_T$ with a detectable, tree-ring-like pattern $m$ in the frequency domain before generating an image. During detection, they leverage the property of LDM's that the diffusion process is invertible, which allows mapping an image back to its original noise. The verification extracts a message $m'$ by spectral analysis and tests whether the same tree-ring patterns $m$ are retained in the frequency domain of the reconstructed noise.

---

[1]Uchida et al. (2017) study watermarking image classifiers. Our categorization is independent of the task.
[2]Embedding can alter the entire generation process, including adding pre- and post-processors.

**Surveyed Watermarking Methods.** In this paper, we evaluate the robustness of five watermarking methods: TRW, WDM, DWT, DWT-SVD, and RivaGAN. DWT, DWT-SVD, and RivaGAN are default choices when using StabilityAI's Stable Diffusion repository and WDM and TRW are two recently proposed methods for Stable Diffusion models. However, WDM requires re-training a Stable Diffusion model from scratch, which can require 150-1000 GPU days (Dhariwal & Nichol, 2021) and is not replicable with limited resources. For this reason, instead of using the autoencoder on the input data, we apply their autoencoder as a post-processor after generating images.

# 3 THREAT MODEL

We consider a provider capable of training large image generators who make their generators accessible to many users via a black-box API, such as OpenAI with DALL·E. Users can query the generator by including a textual prompt that controls the content of the generated image. We consider an attack by an untrustworthy user who wants to misuse the provided generator without detection.

**Provider's Capabilities and Goals** *(Model Capabilities)* The provider fully controls image generation, including the ability to post-process generated images. *(Watermark Verification)* In a no-box setting, the defender must verify their watermark using a single generated image. The defender aims for an effective watermark that preserves generator quality while preventing the attacker from evading detection without significant image quality degradation.

**Attacker's Capabilities.** *(Model Capabilities)* The user has black-box query access to the provider's watermarked model and also has white-box access to less capable, open-source *surrogate* generators, such as Stable Diffusion on Huggingface. We assume the surrogate model's image quality is inferior to the provided model; otherwise, there would be no need to use the watermarked model. Our attacker does not require access to image generators from other providers, but, of course, such access may imply access to surrogate models as our attack does require. *(Data Access)* The attacker has unrestricted access to real-world image and caption data available online, such as LAION-5B (Schuhmann et al., 2022). *(Resources)* Computational resources are limited, preventing the attacker from training their own image generator from scratch. *(Queries)* The provider charges the attacker per image query, limiting the number of queries they can make. The attacker can generate images either unconditionally or with textual prompts. *(Adaptive)* The attacker knows the watermarking method but lacks access to the secret watermarking key $\tau$ and chosen message $m$.

**Attacker's Goal.** The attacker wants to use the provided, watermarked generator to synthesize images (i) without a watermark that (ii) have a high quality. We measure quality using a perceptual similarity function $Q : \mathcal{X} \times \mathcal{X} \to \mathbb{R}$ between the generated, watermarked image and a perturbed image after the attacker evades watermark detection. We require that the defender verifies the presence of a watermark correctly with a p-value of at least $p < 0.01$, same as Wen et al. (2023).

# 4 CONCEPTUAL APPROACH

As described in Section 2, a watermarking method defines three procedures (`KeyGen`, `Embed`, `Verify`). The provider generates a secret watermarking key $\tau \leftarrow \text{KEYGEN}(\theta_G)$ that allows them to watermark their generator so that all its generated images retain the watermark. To embed a watermark, the provider chooses a message (we sample a message $m \sim \mathcal{M}$ uniformly at random) and modifies their generator's parameters $\theta_G^* \leftarrow \text{EMBED}(\theta_G, \tau, m)$. Any image generated by $\theta_G^*$ should retain the watermark. For any $x \leftarrow \text{GENERATE}(\theta_G^*)$ we call a watermark *effective* if (i) the watermark is retained, i.e., $\text{VERIFY}(x, \tau, m) < 0.01$ and (ii) the watermarked images have a high perceptual quality. The attacker generates images $x \leftarrow \text{GENERATE}(\theta_G^*)$ and applies an image-to-image transformation, $\mathcal{A} : \mathcal{X} \to \mathcal{X}$ with parameters $\theta_\mathcal{A}$ to evade watermark detection by perturbing $\hat{x} \leftarrow \mathcal{A}(x)$. Finally, the defender verifies the presence of their watermark in $\hat{x}$, as shown in Figure 1.

Let $W(\theta_G, \tau', m) = \text{EMBED}(\theta_G, \tau', m)$ be the watermarked generator after embedding with key $\tau'$ and message $m$ and $G_W = \text{GENERATE}(W(\theta_G, \tau', m))$ denotes the generation of an image using the watermarked generator parameters. For any high-quality $G_W$, the attacker's objective becomes:

$$\max_{\theta_\mathcal{A}} \mathop{\mathbb{E}}_{\substack{\tau' \leftarrow \text{KEYGEN}(\theta_G) \\ m \in \mathcal{M}}} \left[ \text{VERIFY}(\mathcal{A}(G_W), \tau', m) + Q(\mathcal{A}(G_W), G_W) \right] \tag{2}$$

This objective seeks to maximize (i) the expectation of successful watermark evasion over all potential watermarking keys $\tau'$ and messages $m$ (since the attacker does not know which key-message pair was chosen) and (ii) the perceptual similarity of the images before and after the attack. Note that in this paper, we define image quality as the perceptual similarity to the watermarked image before the attack. There are two obstacles for an attacker to optimize this objective: (1) The attacker has imperfect information about the optimization problem and must substitute the defender's image generator with a less capable, open-source surrogate generator. When KEYGEN depends on $\theta_G$, then the distribution of keys differs, and the attack's effectiveness must transfer to keys generated using $\theta_G$. (2) The optimization problem might be hard to approximate, even when perfect information is available, e.g., when the watermark verification procedure is not differentiable.

## 4.1 MAKING WATERMARKING KEYS DIFFERENTIABLE

We overcome the two aforementioned limitations by (1) giving the attacker access to a similar (but less capable) surrogate generator $\hat{\theta}_G$, enabling them to generate surrogate watermarking keys, and (2) by creating a method GKEYGEN($\hat{\theta}_G$) that creates a surrogate watermarking key $\theta_K$ through which we can backpropagate gradients. A simple but computationally expensive method of creating differentiable keys $\theta_D$ is using Algorithm 1 to train a watermark extraction neural network with parameters $\theta_D$ to predict the message $m$ from an image.

---

**Algorithm 1** GKEYGEN: A Simple Method to Generate Differentiable Keys

---

**Require:** Surrogate generator $\hat{\theta}_G$, Watermarking method (`KeyGen`, `Embed`, `Verify`), $N$ steps
1: $\tau \leftarrow$ KEYGEN($\hat{\theta}_G$)                                   ▷ The surrogate key
2: **for** $j \leftarrow 1$ **to** $N$ **do**
3:      $m \sim \mathcal{M}$                                   ▷ Sample a random message
4:      $\hat{\theta^*_G} \leftarrow$ EMBED($\hat{\theta}_G, \tau, m$)                       ▷ Embed the watermark
5:      $x \leftarrow$ GENERATE($\hat{\theta^*_G}$)
6:      $m' \leftarrow$ EXTRACT($x; \theta_D$)
7:      $g_{\theta_D} \leftarrow \nabla_{\theta_D} ||m - m'||_1$      ▷ Compute gradients using distance between messages
8:      $\theta_D \leftarrow \theta_D - \text{Adam}(\theta_D, g_{\theta_D})$
9: **return** $\theta_D$                                      ▷ The surrogate key

---

Algorithm 1 generates a surrogate key (line 1) to embed a watermark into the surrogate generator and use it to generate watermarked images (lines 3-5). The attacker extracts the message (line 6) and updates the parameters of the (differentiable) watermark decoder using an Adam optimizer (Kingma & Ba, 2014). The attacker subsequently uses the decoder's parameters $\theta_D$ as inputs to VERIFY. Our adaptive attacker must invoke Algorithm 1 only for the non-differentiable watermarks DCT and DCT-SVD (Cox et al., 2007). The remaining three watermarking methods TRW (Wen et al., 2023), WDM (Zhao et al., 2023) and RivaGAN (Zhang et al., 2019) do not require invoking GKEYGEN. In our work, we tune the parameters $\theta_D$ of a ResNet-50 decoder (see Appendix A.3 for details).

## 4.2 LEVERAGING OPTIMIZATION AGAINST WATERMARKS

Equation (2) requires finding attack parameters $\theta_{\mathcal{A}}$ against any watermarking key $\tau' \leftarrow$ KEYGEN($\theta_G$), which can be computationally expensive if the attacker has to invocate GKEYGEN many times. We find empirically that generating many keys is unnecessary, and the attacker can find effective attacks using only a single surrogate watermarking key $\theta_D \leftarrow$ GKEYGEN($\hat{\theta}_G$).

We propose two learnable attacks $\mathcal{A}_1, \mathcal{A}_2$ whose parameters $\theta_{\mathcal{A}_1}, \theta_{\mathcal{A}_2}$ can be optimized efficiently. The first attack, called *Adversarial Noising*, finds adversarial examples given an image $x$ using the surrogate key as a reward model. The second attack called *Adversarial Compression*, first fine-tunes the parameters of a pre-trained autoencoder in a preparation stage and uses the optimized parameters during an attack. The availability of a pre-trained autoencoder is a realistic assumption if the attacker has access to a surrogate Stable Diffusion generator, as the autoencoder is a detachable component of any Stable Diffusion generator. Access to a surrogate generator implies the availability of a pre-trained autoencoder at no additional cost in computational resources for the attacker.

| **Algorithm 2** Adversarial Noising | **Algorithm 3** Adversarial Compression |
|---|---|
| **Require:** surrogate $\hat{\theta}_G$, budget $\epsilon$, image $x$ | **Require:** surrogate $\hat{\theta}_G$, strength $\alpha$, image $x$ |
| 1: $\theta_{\mathcal{A}} \leftarrow 0$ ▷ adversarial perturbation | 1: $\theta_{\mathcal{A}} \leftarrow [\theta_{\mathcal{E}}, \theta_{\mathcal{D}}]$ ▷ Compressor parameters |
| 2: $\theta_D \leftarrow \text{GKEYGEN}(\hat{\theta}_G)$ | 2: $\theta_D \leftarrow \text{GKEYGEN}(\hat{\theta}_G)$ ▷ surrogate key |
| 3: $m \leftarrow \text{EXTRACT}(x; \theta_D)$ | 3: **for** $j \leftarrow 1$ **to** $N$ **do** |
| 4: **for** $j \leftarrow 1$ **to** $N$ **do** | 4: $\quad m \sim \mathcal{M}$ |
| 5: $\quad m' \leftarrow \text{EXTRACT}(x + \theta_{\mathcal{A}}, \theta_D)$ | 5: $\quad \hat{\theta}_G^* \leftarrow \text{EMBED}(\hat{\theta}_G, \theta_D, m)$ |
| 6: $\quad g_{\theta_{\mathcal{A}}} \leftarrow -\nabla_{\theta_{\mathcal{A}}} \|m - m'\|_1$ | 6: $\quad x \leftarrow \text{GENERATE}(\hat{\theta}_G^*)$ |
| 7: $\quad \theta_{\mathcal{A}} \leftarrow P_\epsilon(\theta_{\mathcal{A}} - \text{Adam}(\theta_{\mathcal{A}}, g_{\theta_{\mathcal{A}}}))$ | 7: $\quad x' \leftarrow \mathcal{D}(\mathcal{E}(x; \theta_{\mathcal{A}}))$ ▷ compression |
| $\quad$ **return** $x + \theta_{\mathcal{A}}$ | 8: $\quad m' \leftarrow \text{EXTRACT}(x', \theta_D)$ |
| | 9: $\quad g_{\theta_{\mathcal{A}}} \leftarrow \nabla_\delta(\mathcal{L}_{\text{LPIPS}}(x', x) - \alpha\|m - m'\|_1)$ |
| | 10: $\quad \theta_{\mathcal{A}} \leftarrow \theta_{\mathcal{A}} - \text{Adam}(\theta_{\mathcal{A}}, g_{\theta_{\mathcal{A}}})$ |
| | $\quad$ **return** $\mathcal{D}(\mathcal{E}(x; \theta_{\mathcal{A}}))$ |

**Adversarial Noising**. Algorithm 2 shows the pseudocode of our adversarial noising attack. Given a surrogate generator $\hat{\theta}_G$, a budget $\epsilon \in \mathbb{R}^+$ for the maximum allowed noise perturbation, and a watermarked image $x$ generated using the provider's watermarked model, the attacker wants to compute a perturbation within an $\epsilon$-ball of the $L_\infty$ norm that evades watermark detection. The attacker generates a local surrogate watermarking key (line 2) and extracts a message $m$ from $x$ (line 3). Then, the attacker computes the adversarial perturbation by maximizing the distance to the initially extracted message $m$ while clipping the perturbation into an $\epsilon$-ball using $P_\epsilon$ (line 7).

**Adversarial Compression.** Algorithm 3 shows the pseudocode of our adversarial compression attack. After generating a surrogate watermarking key (line 2), the attacker generates images containing a random message (lines 4-6) and uses their encoder-decoder pair to compress the images (line 7). The attacker iteratively updates their model's parameters by (i) minimizing a quality loss, which we set to the LPIPS metric (Zhang et al., 2018), and (ii) maximizing the distance between the extracted and embedded messages (line 9). The output $\theta_{\mathcal{A}}$ of the optimization loop between lines 3 and 10 only needs to be run once, and the weights $\theta_{\mathcal{A}}$ can be re-used in subsequent attacks.

We highlight that the attacker optimizes an approximation of Equation (2) since they only have access to a surrogate generator $\hat{\theta}_G$, but not the provider's generator $\theta_G$. This may lead to a generalization gap of the attack at inference time. Even if an attacker can find optimal attack parameters $\theta_{\mathcal{A}}$ that optimizes Equation (2) using $\hat{\theta}_G$, the attacker cannot test whether their attack remains effective when the defender uses a different model $\theta_G$ to generate watermarking keys.

## 5 EXPERIMENTS

**Image Generators.** We experiment with Stable Diffusion, an open-source, state-of-the-art latent diffusion model. The defender deploys a Stable Diffusion-v2.0 model[3] trained for 1.4m steps in total on a subset of LAION-5B (Schuhmann et al., 2022). The attacker uses a less capable Stable Diffusion-v1.1[4] checkpoint, trained for 431k steps in total on LAION-2B and LAION-HD. All experiments were conducted on NVIDIA A100 GPUs.

Images are generated using a DPM solver (Lu et al., 2022) with 20 inference steps and a default guidance scale of 7.5. We create three different watermarked generators for each surveyed watermarking method by randomly sampling a watermarking key $\tau \leftarrow \text{KEYGEN}(\theta_G)$ and a message $m \sim \mathcal{M}$, used to embed a watermark. Appendix A.1 contains descriptions of the watermarking keys. All reported values represent the mean value over three independently generated secret keys.

**Quantitative Analysis.** Similar to Wen et al. (2023), we report the True Positive Rate when the False Positive Rate is fixed to 1%, called the TPR@1%FPR. Appendix A.2 describes statistical tests used in the verification procedure of each watermarking method to derive p-values. We report the

---

[3] https://huggingface.co/stabilityai/stable-diffusion-2-base
[4] https://huggingface.co/CompVis/stable-diffusion-v1-1

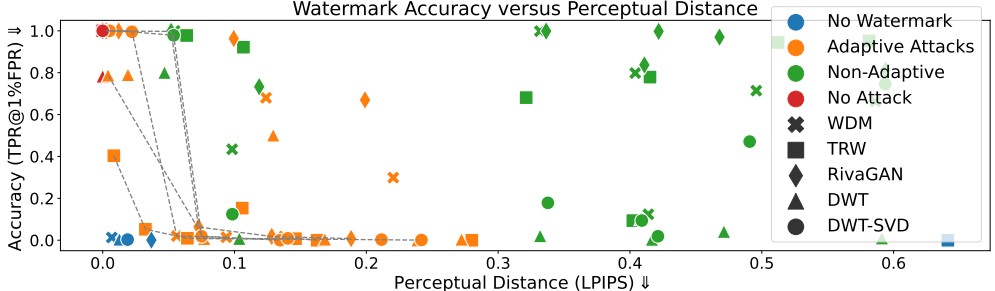

Figure 2: The effectiveness of our attacks against all watermarks. We highlight the Pareto front for each watermarking method by dashed lines and indicate adaptive/non-adaptive attacks by colors.

Fréchet Inception Distance (FID) (Heusel et al., 2017), which measures the similarity between real and generated images. Additionally, we report the CLIP score (Radford et al., 2021) that measures the similarity of a prompt to an image. We generate 1k images to evaluate TPR@1%FPR and 5k images to evaluate FID and CLIP score on the training dataset of MS-COCO-2017 (Lin et al., 2014).

## 5.1 EVALUATING ROBUSTNESS

Figure 2 shows a scatter plot of the effectiveness of our attacks against all surveyed watermarking methods. We evaluate adaptive and non-adaptive attacks. Similar to Wen et al. (2023), for the non-adaptive attacks, we use Blurring, JPEG Compression, Cropping, Gaussian noise, Jittering, Quantization, and Rotation but find these attacks to be ineffective at removing the watermark. Figure 2 highlights Pareto optimal attacks for pairs of (i) watermark detection accuracies and (ii) perceptual distances. We find that only adaptive attacks evade watermark detection and preserve image quality.

Table 1 summarizes the best attacks from Figure 2 when we set the lowest acceptable detection accuracy to $10\%$. When multiple attacks achieve a detection accuracy lower than $10\%$, we pick the attack with the lowest perceptual distance to the watermarked image. We observe that adversarial compression is an effective attack against all watermarking methods. TRW is also evaded by adversarial compression, but adversarial noising at $\epsilon = 2/255$ preserves a higher image quality.

|  | TRW | WDM | DWT | DWT-SVD | RivaGAN |
|---|---|---|---|---|---|
| Best Attack | Adv. Noising | Compression | Compression | Compression | Compression |
| Parameters | $\epsilon = 2/255$ | $r = 1$ | $r = 1$ | $r = 1$ | $r = 1$ |
| LPIPS $\Downarrow$ | 3.2e-2 | 5.6e-2 | 7.7e-2 | 7.5e-2 | 7.3e-2 |
| Accuracy $\Downarrow$ | 5.2% | 2.0% | 0.8% | 1.9% | 6.3% |

Table 1: A summary of Pareto optimal attacks with detection accuracies less than $10\%$. We list the attack's name and parameters, the perceptual distance before and after evasion, and the accuracy (TPR@1%FPR). $\epsilon$ is the maximal perturbation in the $L_\infty$ norm and $r$ is the number of compressions.

## 5.2 IMAGE QUALITY AFTER AN ATTACK

Figure 3 shows the perceptual quality after using our adaptive attacks. We show a cutout of the top left image patch with high contrasts on the bottom right to visualize noise artifacts potentially introduced by our attacks. We observe that, unlike adversarial noising, the compression attack introduces no new visible artifacts (see also Appendix A.4 for more visualizations).

The FID and CLIP scores of the watermarked images and the images after using adversarial noising and adversarial compression remain unchanged (see Table 2 in the Appendix). We calculate the quality using the best attack configuration from Figure 2 when the detection accuracy is less than 10%. Adversarial Noising is ineffective at removing WDM and RivaGAN for $\epsilon \leq 10/255$.

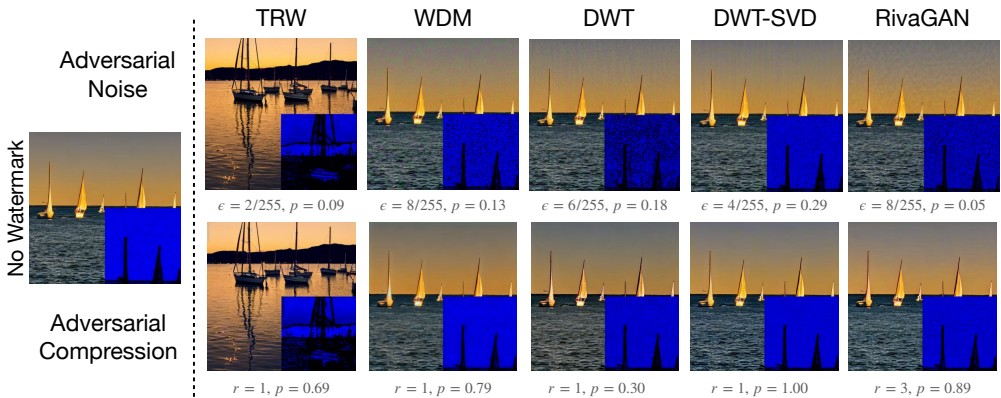

Figure 3: A visual analysis of two adaptive attacks. The left image shows the unwatermarked output, including a high-contrast cutout of the top left corner of the image to visualize noise artifacts. On the right are images after evasion with adversarial noising (top) and adversarial compression (bottom).

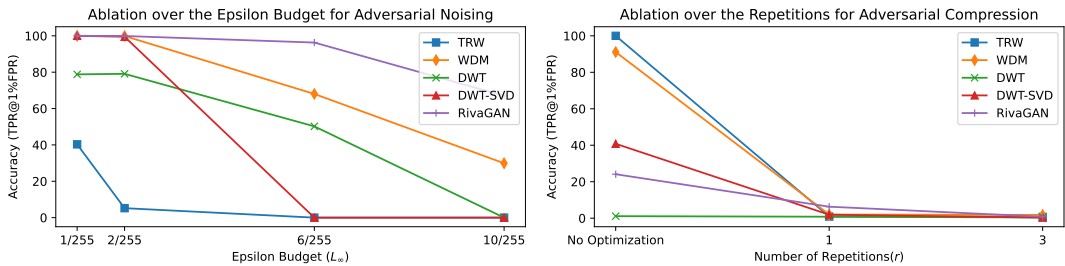

Figure 4: Ablation studies over (left) the maximum perturbation budget $\epsilon$ in $L_\infty$ for adversarial noising and (right) the number of adversarial compressions against each watermarking method. "No Optimizations" means we did not optimize the parameters $\theta_\mathcal{A}$ of the attack.

## 5.3 ABLATION STUDY

Figure 4 shows ablation studies for our adaptive attacks over the (i) maximum perturbation budget and (ii) the number of compressions applied during the attack. TRW and DWT-SVD are highly vulnerable to adversarial noising, whereas RivaGAN and WDM are substantially more robust to these types of attacks. We believe this is because keys generated by RivaGAN and WDM are sufficiently randomized, which makes our attack (that uses only a single surrogate key) less effective unless the surrogate key uses similar channels as the secret key to hide the watermark. Adversarial compression *without* optimization of the parameters $\theta_\mathcal{A}$ is ineffective at evading watermark detection against all methods except DWT. After optimization, adversarial compression evades detection from all watermarking methods with only a single compression.

## 6 DISCUSSION & RELATED WORK

**Attack Scalability.** The presented findings clearly indicate that even with a less capable surrogate generator, an adaptive attacker can remove all surveyed watermarks with minimal quality degradation. Our attackers generate a single surrogate key and are able to evade watermark verification, which indicates a design flaw since the key seems to have little impact. If KEYGEN were sufficiently randomized, breaking robustness should not be possible using a single key, even if the provided and surrogate generators are the same. An interesting question emerging from our study relates to the maximum difference between the watermarked and surrogate generators for the attacks to remain effective. We used a best-effort approach, by using two public checkpoints with the largest reported quality differences: Stable Diffusion v1.1 and v2. More research is needed to study the impact on the effectiveness of our attacks (i) using different models (ii) or limiting the attacker's knowledge of the method's public parameters. Our attacks evaluate the best parameters suggested by the authors.

**Types of Learnable Attacks.** Measuring the robustness against different types of learnable attacks is crucial in assessing the trustworthiness of watermarking. We explored (i) Adversarial Examples, which rely solely on the surrogate key, and (ii) Adversarial Compression, which additionally requires the availability of a pre-trained autoencoder. We believe this requirement is satisfied in practice, given that (i) training autoencoders is computationally less demanding than training Stable Diffusion, and many pre-trained autoencoders have already been made publicly available (Podell et al., 2023). Although autoencoders enhance an attacker's ability to modify images, our study did not extend to other learnable attacks such as inpainting (Rombach et al., 2022) or more potent image editing methods (Brooks et al., 2023) which could further enhance an attack's effectiveness.

**Enhancing Robustness using Adaptive and Learnable Attacks**. Relying on non-adaptive attacks for evaluating a watermark's robustness is inadequate as it underestimates the attacker's capabilities. To claim robustness, the defender could (i) provide a certification of robustness (Bansal et al., 2022), or (ii) showcase empirically that their watermark withstands strong attacks. The issue is that we lack strong attackers. Although Lukas et al. (2022) demonstrated that adaptive attackers can break watermarks for image classifiers, their attacks were handcrafted and did not scale. Instead, we propose a better method of empirically testing robustness by proposing adaptive *learnable* attackers that require only the specification of a type of learnable attack, followed by an optimization procedure to find parameters that minimize an objective function. We believe that any watermarking method proposed in the future should evaluate robustness using our attacks and expect that future watermarking methods can enhance their robustness by incorporating our attacks.

**Limitations.** Our attacks are based on the availability of the watermarking algorithm and an open-source surrogate generator to replicate keys. While providers like Stable Diffusion openly share their models, and replicas of OpenAI's DALL·E models are publicly available (Dayma et al., 2021), not all providers release information about their models. To the best of our knowledge, Google has not released their generators (Saharia et al., 2022), but efforts to replicate are ongoing[5]. Providers like Midjourney, who keep their image generation algorithms undisclosed, prevent adaptive attackers altogether but may be vulnerable to these attacks by anyone to whom this information is released.

**Outlook.** An adaptive attacker can instantiate more effective versions of their attacks with knowledge of the watermarking method's algorithmic descriptions (KEYGEN, EMBED, VERIFY). Our attacks require no interaction with the provider. Robustness against adaptive attacks extends to robustness against non-adaptive attacks, which makes studying the former interesting. Adaptive attacks will remain a useful tool for studying a watermarking method's robustness, even if the watermarking method is kept secret. Previous works did not consider such attacks, and we show that future watermarking methods must consider them if they claim robustness empirically.

## 6.1 RELATED WORK

Jiang et al. (2023) propose attacks that use (indirect) access to the secret watermarking key via access to the provider's VERIFY method. Our attacks require no access to the provider's secret watermarking key, as our attacks optimize over any key message pair (see Equation (2)). These threats are related since both undermine robustness but are orthogonal due to different threat models. Peng et al. (2023); Cui et al. (2023) propose black-box watermarking methods that protect the Intellectual Property of Diffusion Models. We focus on no-box verifiable watermarking methods that control misuse. Lukas & Kerschbaum (2023); Yu et al. (2020; 2021) propose watermarking methods but only evaluate GANs. We focus on watermarking methods for pre-trained Stable Diffusion models with much higher output diversity and image quality (Dhariwal & Nichol, 2021).

## 7 CONCLUSION

We propose testing the robustness of watermarking through adaptive, learnable attacks. Our empirical analysis shows that such attackers can evade watermark detection against all five surveyed image watermarks. Adversarial noising evades TRW (Wen et al., 2023) with $\epsilon = 2/255$ but needs to add visible noise to evade the remaining four watermarking methods. Adversarial compression evades all five watermarking methods using only a single compression. We encourage using these adaptive attacks to test the robustness of watermarking methods in the future more comprehensively.

---

[5] https://github.com/lucidrains/imagen-pytorch

## 8 ETHICS STATEMENT

The attacks we provide target academic systems, and the engineering efforts to attack real systems are substantial. We make it harder by not releasing our code publicly. We will, however, release our code, including pre-trained checkpoints, upon carefully considering each request. Currently, there are no known security impacts of our attacks since users cannot yet rely on the provider's use of watermarking. The use of watermarking is experimental and occurs at the provider's own risk, and our research aims to improve the trustworthiness of image watermarking by evaluating it more comprehensively.

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

## A   APPENDIX

### A.1   PARAMETERS FOR WATERMARKING METHODS

**Tree Ring Watermark (TRW)** (Wen et al., 2023): We evaluate the Tree-Ring$_{\text{Rings}}$ method, which the authors state "delivers the best average performance while offering the model owner the flexibility of multiple different random keys". Using the author's implementation[6], we generate and verify watermarks using 20 inference steps, where we use no knowledge of the prompt during verification and keep the remaining default parameters chosen by the authors.

**Watermark Diffusion Model (WDM)** (Zhao et al., 2023): As stated in the paper, instead of stamping the model's training data to embed a watermark, we apply the pre-trained encoder to a generated image as a post-processing step. We choose messages with $n = 40$ bits and use the encoder architecture proposed by (Yu et al., 2021), followed by a ResNet-50 decoder. Each call to KEYGEN($\hat{\theta}_G$), where $\hat{\theta}_G$ is the surrogate generator, trains a new autoencoder from scratch.

**DWT, DWT-SVD** (Cox et al., 2007) and **RivaGAN** (Zhang et al., 2019). We use 32-bit messages and keep the default parameters set in the implementation used by the Stable Diffusion models[7].

### A.2   STATISTICAL TESTS

**Matching Bits.**   WDM (Zhao et al., 2023), DWT, DWT-SVD (Cox et al., 2007) and Riva-GAN (Zhang et al., 2019) encode messages $m \in \mathcal{M}$ by bits and our goal is to verify whether

---

[6]`https://github.com/YuxinWenRick/tree-ring-watermark`
[7]`https://github.com/ShieldMnt/invisible-watermark`

message $m \in \mathcal{M}$ is present in $x \in \mathcal{X}$ using key $\tau$. We extract $m'$ from $x$ and want to reject the following null hypothesis.

$$H_0 : m \text{ and } m' \text{ match by random chance.}$$

For a given pair of bit-strings of length $n$, if we denote the number of matching bits as $k$, the expected number of matches by random chance follows a binomial distribution with parameters $n$ and expected value $0.5$. The p-value for observing at least $k$ matches is given by:

$$p = 1 - \text{CDF}(k - 1; n, 0.5) \tag{3}$$

Where CDF represents the cumulative distribution function of the binomial distribution.

**Matching Latents.** TRW (Wen et al., 2023) leverages the forward diffusion process of the diffusion model to reverse an image $x$ to its initial noise representation $x_T$. This transformation is represented by $m' = \mathcal{F}(x_T)$, where $\mathcal{F}$ denotes a Fourier transform. The authors find that reversed real images and their representations in the Fourier domain are expected to follow a Gaussian distribution. The watermark verification process aims to reject the following null hypothesis:

$$H_0 : y \text{ originates from a Gaussian distribution } N(0, \sigma_{IC}^2)$$

Here, $y$ is a subset of $m'$ based on a watermarking mask chosen by the provider, which determines the relevant coefficients. The test statistic, $\eta$, denotes the normalized sum-of-squares difference between the original embedded message $m$ and the extracted message $m'$, which can be complex-valued due to the Fourier transform. Specifically,

$$\eta = \frac{1}{\sigma^2} \sum_i |m_i - m_i'|^2 \tag{4}$$

And,

$$p = \text{Pr}\left(\chi_{|M|,\lambda}^2 \leq \eta \mid H_0\right) = \Phi_{\chi^2}(\eta) \tag{5}$$

Where $\Phi_{\chi^2}$ represents the cumulative distribution function of the noncentral $\chi^2$ distribution. We refer to Wen et al. (2023) for more detailed descriptions of these statistical tests.

## A.3 DETAILS ON GKEYGEN

This section provides more details on Algorithm 1. VERIFY consists of a sub-procedure EXTRACT, which maps an image to a message using the secret key, as stated in Section 2.1. The space of messages is specific to the watermarking method. We consider two message spaces $\mathcal{M}$: multi-bit messages and messages in the Fourier space. All surveyed methods except TRW are multi-bit watermarks, for which we use the categorical cross-entropy to measure similarity, and for TRW, we use the mean absolute error as a similarity measure between messages (line 7 of Algorithm 1).

As stated in the main paper, we instantiate GKEYGEN only for the DWT and DWT-SVD watermarking methods. We train a ResNet-50 decoder $\theta_D$ in Algorithm 1 to predict a bit vector of the same length as the message and calculate gradients during training using the cross-entropy loss. Attacking TRW, WDM, and RivaGAN only requires invoking KEYGEN, as the keys are the parameters of (differentiable) decoders. We train these keys from scratch for WDM and RivaGAN. For TRW, the key generation does not require any training, as the key only specifies elements in the Fourier space that encode the message. The forward diffusion process used in VERIFY is already differentiable.

## A.4 QUALITATIVE ANALYSIS OF WATERMARKING TECHNIQUES

We refer to Figure 5 for examples of non-watermarked, watermarked, and attacked images using the attacks summarized in Table 1. We show three images for each of the five surveyed watermarking methods: an image without a watermark, one with a watermark, and the watermarked image after an evasion attack. We show the prompt that was used to generate these images and label each image with the p-value with which the expected message was detected in the image using the secret watermarking key and the VERIFY procedure.

A.5 QUALITY EVALUATION

Table 2 shows the FID and CLIPScore of all five surveyed watermarking methods without a watermark (first row), with a watermark (second row), after our adaptive noising attack (third row) and after our adversarial compression attack (fourth row). All results are reported as the mean value over three independent runs using three different secret watermarking keys. We observe that the degradation in FID and CLIPScores is statistically insignificant, as seen in Figure 5.

|  | TRW | | WDM | | DWT | | DWT-SVD | | RivaGAN | |
|---|---|---|---|---|---|---|---|---|---|---|
|  | FID | CLIP | FID | CLIP | FID | CLIP | FID | CLIP | FID | CLIP |
| No WM | 23.32 | 31.76 | 23.48 | 31.77 | 23.48 | 31.77 | 23.48 | 31.77 | 23.48 | 31.77 |
| WM | 24.19 | 31.78 | 23.43 | 31.72 | 23.16 | 32.11 | 23.10 | 32.15 | 22.96 | 31.84 |
| A-Noise | 23.67 | 32.15 | N/A | N/A | 23.55 | 32.46 | 22.89 | 32.50 | N/A | N/A |
| A-Comp | 24.36 | 31.87 | 23.27 | 32.01 | 23.16 | 32.17 | 23.06 | 31.92 | 23.25 | 31.86 |

Table 2: Quality metrics before and after watermark evasion. FID$\Downarrow$ represents the Fréchet Inception Distance, and CLIP$\Uparrow$ represents the CLIP score, computed on 5k images from MS-COCO-2017. N/A means the attack could not evade watermark detection, and we do not report quality measures.

A.6 ATTACK EFFICIENCY

From a computational perspective, generating a surrogate watermarking key with methods such as RivaGAN or WDM is the most expensive operation, as it requires training a watermark encoder-decoder pair from scratch. Generating a key for these two methods takes around 4 GPU hours each on a single A100 GPU, which is still negligible considering the total training time of the diffusion model, which takes approximately 150-1000 GPU days (Dhariwal & Nichol, 2021). The optimization of Adversarial noising takes less than 1 second per sample, and tuning the adversarial compressor's parameters takes less than 10 minutes on a single A100 GPU.

A.7 DOUBLE-TAIL DETECTION

Jiang et al. (2023) propose a more robust statistical test that uses two-tailed detection for multi-bit messages. The idea is to test for the presence of a watermark with message $m$ or message $1 - m$ (all bits flipped). We implemented the double-tail detection described by Jiang et al. (2023) and adjusted the statistical test in VERIFY to use double-tail detection on the same images used in Figure 4. Table 3 summarizes the resulting TPR@1%FPR with single or double-tail detection. Since TRW (Wen et al., 2023) is not a multi-bit watermarking method, we omit its results.

| Attack Method | WDM | DWT | DWT-SVD | RivaGAN |
|---|---|---|---|---|
| Adv. Noising ($\varepsilon = 1/255$) | 100% / 100% | 78.8% / 75.9% | 100% / 100% | 100% / 100% |
| Adv. Noising ($\varepsilon = 2/255$) | 99.9% / 99.0% | 79.1% / 75.0% | 99.5% / 99.0% | 99.9% / 99.9% |
| Adv. Noising ($\varepsilon = 6/255$) | 68.0% / 68.7% | 50.2% / 49.5% | **0.0%** / 23.3% | 96.3% / 96.6% |
| Adv. Noising ($\varepsilon = 10/255$) | 29.9% / 36.9% | **0.0%** / 57.3% | **0.0%** / 11.8% | 67.0% / 63.3% |
| Adv. Compression | **2.0% / 1.8%** | **0.8% / 0.5%** | **1.9% / 2.5%** | **6.3% / 5.7%** |

Table 3: Summary of TPR@1%FPR using single-tail (left) and double-tail detection (right). We mark attacks bold if their TPR@1%FPR is less than 10%.

Table 3 shows that double-tail detection increases the robustness of DWT and DWT-SVD against adversarial noising, which is the same effect that Jiang et al. (2023) find in their paper. We find that Adversarial Compression remains effective against all attacks in the presence of double-tail detection. Figure 4 shows that adversarial noising is highly effective against TRW but is ineffective against the remaining methods because an attacker has to add visible noise (see Figure 3). An attacker would always use the Adversarial Compression attack in a real-world attack.

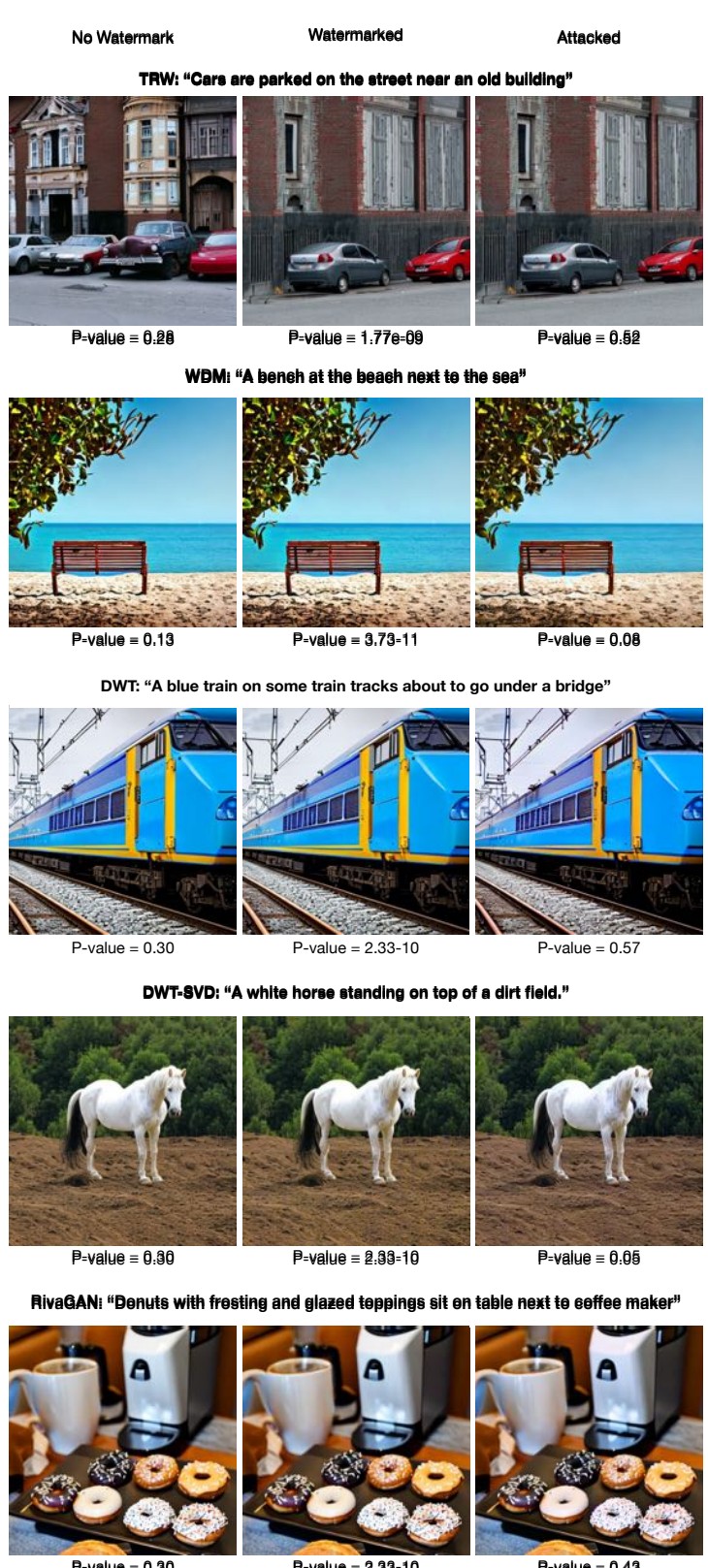

Figure 5: Qualitative showcase of three kinds of images: non-watermarked, watermarked with mentioned technique, and attacked images with the strongest attack from Table 1. The p-values and text prompts are also provided.

