# OpenReview forum: "Leveraging Optimization for Adaptive Attacks on Image Watermarks"
_ICLR.cc/2024/Conference — ICLR 2024 poster_

### Official Review · Reviewer_2NFv · 2023-10-30

**Soundness:** 3 good
**Presentation:** 3 good
**Contribution:** 3 good
**Rating:** 6
**Confidence:** 3

**Summary:**

This paper introduces an objective function and approaches adaptive attacks as an optimization problem to evaluate the robustness of watermarking algorithms.  The core idea of the proposed adaptive attack is to replicate secret watermarking keys locally by creating surrogate keys that are differentiable and can be used to optimize the attack’s parameters. The experiments reveal that this type of attacker can successfully compromise all five surveyed watermarking methods with minimal degradation in image quality. These findings underscore the necessity for more comprehensive testing of the robustness of watermarking algorithms against adaptive, learnable adversaries.

**Strengths:**

This paper proposes a practical method of empirically testing the robustness of different watermarking methods. The proposed adaptive attack is shown to be effective against different watermarking methods.

**Weaknesses:**

1. The attack requires the knowledge of the watermarking algorithm.
2 in the experiments the surrogate generator and the watermark generator exhibited a high degree of similarity, which may not be as practical in real-world scenarios.

**Questions:**

1 The paper shows that generates a single surrogate key and can evade watermark verification which indicates that the private key seems to have little impact. Is it possible that the less impact comes from the high similarity between the surrogate generator and the watermark generator?

---

> ### Author Response · Authors · 2023-11-13
> **Thank you for your comments!**
>
> We thank the reviewer for their comments and positive assessment of our paper. Please find detailed responses below.
>
> > The attack requires the knowledge of the watermarking algorithm.
>
> Knowledge of the watermarking algorithm may only sometimes be given in a real-world attack (as outlined in our limitations on page 9). Still, it remains a valuable tool for the provider to _test_ the robustness of a watermarking method. Robustness against adaptive attackers extends to robustness against non-adaptive attackers (as they have fewer capabilities). This forms our core argument for why studying a method’s robustness requires considering adaptive attacks. We will emphasize this argument in the revised paper.
>
> > The paper shows that generates a single surrogate key and can evade watermark verification which indicates that the private key seems to have little impact. Is it possible that the less impact comes from the high similarity between the surrogate generator and the watermark generator?
>
> We thank the reviewer for this interesting question! Our work shows that an attacker can remove any watermark by generating a single surrogate key. If the KEYGEN procedure were sufficiently randomized, this should not be possible even if the attacker had access to **the same** generator as the defender. Our attacks assume less because we use a _surrogate_ generator that is not the same, because the generator’s abuse by the attacker may otherwise not be well motivated, but show that these restricted attacks are still successful with only a single surrogate key. From these results, we conclude that the KeyGen procedure is not sufficiently randomized, which indicates a design flaw. We will add this to the revised paper’s discussion.
>
> We are happy to respond to any further questions or suggestions the reviewer may have.

---

### Official Review · Reviewer_LPhx · 2023-10-31

**Soundness:** 3 good
**Presentation:** 3 good
**Contribution:** 3 good
**Rating:** 8
**Confidence:** 5

**Summary:**

This paper proposes a method for performing adaptive attacks against image watermarking methods, allowing for more accurate evaluations of the robustness of watermarking methods against motivated attackers. To enable standard adversarial optimization attacks against known but non-differentiable watermarking methods, the authors propose to train surrogate (differentiable) watermark detection networks. Experiments show that adaptive attacks crafted with the proposed method significantly degrade the effectiveness of all evaluated watermarking methods and outperform non-adaptive attacks while preserving the overall perceptual quality of attacked images.

**Strengths:**

The issue of watermarking the outputs of generative models is timely and interesting.

The idea of training differentiable surrogates for arbitrary watermarking methods is an interesting threat model.

The selection of baseline watermarking methods is reasonable and includes both "post-hoc" (low-perturbation) and "semantic" (high-perturbation) methods.

The autoencoder/compression-based attack is interesting and seems to effectively remove watermarks while retaining high perceptual quality.

**Weaknesses:**

I think there is a terminology issue in the paper that could be confusing for readers. It appears the watermark "key" referenced in the paper more closely matches the concept of a watermark "detector" algorithm in methods such as RivaGAN and Tree-Rings; many methods often use "key" and "message" interchangeably to refer to the hidden signal. If this is true, the authors' proposed training of differentiable surrogate "keys" can be understood as training differentiable surrogate detector networks that predict the key/message concealed in a watermarked image --  allowing for gradient-based optimization attacks on non-differentiable detectors. The pseudocode in Algorithm 1 strongly suggests this. If this is the case, I urge the authors to revise their terminology to make this more clear.

- - - -

It should be clarified that in the general no-box watermarking scenario, the second step $\mathrm{EMBED}$ need not modify the parameters of the generator model, just endow it with watermarking capabilities (e.g. by applying a post-hoc watermarking algorithm to its outputs). As far as I can tell, none of the methods evaluated in the paper modify the generator parameters directly. Overall, the no-box watermarking steps and their instantiations for each of the evaluated watermarking methods are not clearly explained.

- - - -

The proposed method is similar to that of Jiang et al. [5] in that it adversarially parameterizes image transformations to remove watermarks. The authors claim that the method of Jiang et al. requires access to a watermarking "key;" however, Jiang et al. propose attacks under the explicit assumption that the attacker does not have access to the ground-truth key/message (which is either approximated via the detector model's predictions or sampled at random) -- from pp.6, "the attacker does not have access to the ground-truth watermark $w$".  On the other hand, if I am correct that the authors take "key" to mean "detector," this claim makes more sense in that Jiang et al. use full access (white-box) or query access (black-box) to the detector to craft attacks.

The authors should clarify their statements about the prior work of Jiang et al. and the distinctions between their methods. And while additional experiments may not be feasible at this point for various reasons, I think the paper would be much stronger if it included comparisons between the proposed approach and either or both attack variants proposed by Jiang et al. This would also require modifying the proposed attack to evade two-tailed detection, as discussed by Jiang et al.

- - - -

If the attacker "does not need to invoke GKEYGEN" for TRW/WDM/RivaGAN, as stated in section 4.2, does this mean the attacker does not train a differentiable surrogate detector? In this case, doesn't the proposed approach just become a standard white-box attack, as in Jiang et al.?

- - - -

The authors do not provide any details of the architecture of the surrogate detector networks $\theta_D$ trained by the adversary. This seems like a crucial aspect of the proposed approach, so it is strange that it is not discussed.

- - - -

The related work section mixes references to image classifier watermarks [1][2] and watermarks for generative models [3], which are very different: the former aims to protect the intellectual property of a model developer, typically through query- or trigger-based verification, while the latter is embedded in all outputs of a generative model to distinguish real from fake content. This paper is concerned with the latter kind of watermark, so I'm confused by the emphasis on works in the former area. At the very least, these two different types of works should be clearly distinguished from one another in the related work section.

- - - -

As far as I can tell, the term "Adaptive Attack" comes from the adversarial example literature -- the authors should explain what distinguishes an adaptive attack from a non-adaptive attack and probably cite the original work [4].

- - - -

Figure 1 is missing step #8 (it skips from 7 to 9).

- - - -

This is a much smaller concern, but the substitution of the Stable Diffusion v1 generator for v2 does not seem like a very difficult obstacle for the attacker to overcome, given the general similarities in architecture and training. Attacks on post-hoc methods probably shouldn't be affected too much by the choice of surrogate generator, but Tree-Rings is deeply intertwined with the generator structure. Therefore, it would be interesting to see how attacks on TRW fare when there is a more substantial mismatch between the actual and surrogate generator.

- - - -

Overall, I think the central idea -- no-box watermark attacks with differentiable surrogates -- is very interesting, and the experimental results look very strong. However, I think the paper has many issues that still need to be addressed.


[1] Nils Lukas, Edward Jiang, Xinda Li, and Florian Kerschbaum. Sok: How robust is image classification deep neural network watermarking? In 2022 IEEE Symposium on Security and Privacy
(SP), pp. 787–804. IEEE, 2022.

[2] Arpit Bansal, Ping-yeh Chiang, Michael J Curry, Rajiv Jain, Curtis Wigington, Varun Manjunatha, John P Dickerson, and Tom Goldstein. Certified neural network watermarks with randomized smoothing. In International Conference on Machine Learning, pp. 1450–1465. PMLR, 2022.

[3] Yuxin Wen, John Kirchenbauer, Jonas Geiping, and Tom Goldstein. Tree-ring watermarks: Fingerprints for diffusion images that are invisible and robust. arXiv preprint arXiv:2305.20030, 2023.

[4] Nicholas Carlini and David Wagner. Adversarial Examples Are Not Easily Detected: Bypassing Ten Detection Methods. In AISec '17: Proceedings of the 10th ACM Workshop on Artificial Intelligence and Security, pp.3-14. 2017.

[5] Zhengyuan Jiang, Jinghuai Zhang, and Neil Zhenqiang Gong. Evading watermark based detection of ai-generated content. arXiv preprint arXiv:2305.03807, 2023.

**Questions:**

Did the authors train surrogate detectors for all the evaluated watermarking methods to create true no-box attacks?

Did the authors experiment with different degrees of attacker knowledge -- e.g., what if the attacker does not know the length of the watermark embedded by the actual generator? Would training on 32-bit messages cause attacks on a 64-bit message system to fail?

Rather than training surrogate detectors to reconstruct embedded messages, did the authors consider simply training a binary classifier on watermarked and un-watermarked images from the surrogate generator and then performing an adversarial attack on the binary classifier? This seems like the simplest no-box approach.

---

> ### Author Response · Authors · 2023-11-13
> **Thank you for your comments!**
>
> We appreciate the reviewer’s detailed comments and suggestions for further improvements of our work and will carefully incorporate them in the revised paper. Please find our point-by-point responses below.
>
> > Terminology issue between a "key" and a "message".
>
> Our terminology is the following: A watermarking key refers to secret random bits of information used in the randomized “detector” algorithm (called VERIFY in the paper) to detect a watermark. A message is a hidden signal in the image. This notation is inspired by related works [A, B] that also distinguish the terms “key” and “message" like our work. The Tree-Rings paper instead uses the word “key” to refer to a “message”, which we agree is confusing.
>
> With our terminology, the _key_ in Tree-Rings are the parameters of the diffusion model, the VERIFY function corresponds to the forward diffusion process, and the _messages_ are initial noise vectors for the diffusion process in the frequency domain. We hope that this explanation and example clear up any confusion. We thank the reviewer for bringing this to our attention and will clarify it in the revised paper.
>
> > The proposed method is similar to that of Jiang et al. [5] in that it adversarially parameterizes image transformations to remove watermarks. The authors claim that the method of Jiang et al. requires access to a watermarking "key;" however, Jiang et al. propose attacks under the explicit assumption that the attacker does not have access to the ground-truth key/message [..]. On the other hand, if I am correct that the authors take "key" to mean "detector," this claim makes more sense in that Jiang et al. use full access (white-box) or query access (black-box) to the detector to craft attacks.
> The authors should clarify their statements about the prior work of Jiang et al. and the distinctions between their methods. And while additional experiments may not be feasible at this point for various reasons, I think the paper would be much stronger if it included comparisons between the proposed approach and either or both attack variants proposed by Jiang et al.
>
> As the reviewer correctly points out, there are fundamental differences between our attacks and those proposed by Jiang et al.. The threats studied by Jiang et al. and those studied in our works are related since both undermine robustness but are orthogonal due to different threat models.
>
> **Jiang et al.**: In all attacks, Jiang et al. assume some (at least indirect) access to the secret key (e.g., black-box access to VERIFY, whose response depends on the secret key). They craft an adversarial perturbation that fools the specific instance of the detector for one key-message pair, which is a special case of Eq. 1, where they optimize \theta_A with a single, fixed key \tau and message m (i..e, they remove the expectation term). In their black-box attacks, access to VERIFY is also limited, requiring them to instantiate gradient-free optimization or train surrogate decoders to extract the VERIFY functionality.
>
> **Our work**: Our attacks make no assumptions about the key-message pair used by the provider since they optimize the expected evasion rate over _any_ pair and any instance of the detector. Consequently, Jiang et al. should find “better” attacks (i.e., less visible ones) to evade detection since they make more assumptions about the optimization problem in some regards (optimize against one key-message pair with access to VERIFY) and fewer assumptions in other regards (no knowledge of the watermarking method in the black-box case). Still, unlike our work, their attacks require the attacker to access the provider’s secret key by calling VERIFY.
>
> Both threats are orthogonal because robustness against our attacks does not imply robustness against Jiang et al. 's attacks and vice-versa. To defend against Jiang et al., a provider could restrict their access to the secret key (e.g., allowing only trusted users to access VERIFY or not replying to “similar” queries), but they would remain vulnerable to our attacks. Similarly, a defense against our attack may not defend against Jiang et al.’s attacks when the attacker has additional capabilities, such as access to the provider’s VERIFY procedure. Providers need to defend against both types of attacks independently.
>
> We will highlight the differences between both types of attacks, extend the difference between our work to Jiang et al.’s work, and emphasize the necessity to defend against _both_ threats in the revised paper.
>
> ---------
> [A] Zhao, Xuandong, et al. "Provable robust watermarking for ai-generated text." arXiv preprint arXiv:2306.17439 (2023).
>
> [B] Christ, Miranda, Sam Gunn, and Or Zamir. "Undetectable Watermarks for Language Models." arXiv preprint arXiv:2306.09194 (2023).

---

> > ### Author Response · Authors · 2023-11-13
> > **additonal reply to the comments.**
> >
> > > This would also require modifying the proposed attack to evade two-tailed detection, as discussed by Jiang et al.
> >
> > As far as we can tell, Jiang et al. propose two-tailed detection only for multi-bit messages, but Tree-Rings operates under a different message space; hence, we omit its results. We implemented the double-tail detection described by Jiang et al. and adjusted our VERIFY statistical test to use double-tail detection without re-running the evasion attacks. The table below summarizes the resulting TPR@1%FPR when using single-tail detection (left value) and double-tail detection (right value, in italics) on the same images.
> >
> >
> > | Attack Method           | WDM         | DWT         | DWT-SVD     | RivaGAN     |
> > |-------------------------|-------------|-------------|-------------|-------------|
> > | Adv. Noising (ε=1/255)  | 100% / *100%* | 78.8% / *75.9%* | 100% / *100%* | 100% / *100%* |
> > | Adv. Noising (ε=2/255)  | 99.9% / *99.0%* | 79.1% / *75.0%* | 99.5% / *99.0%* | 99.9% / *99.9%* |
> > | Adv. Noising (ε=6/255)  | 68.0% / *68.7%* | 50.2% / *49.5%* | 0.0% / *23.3%* | 96.3% / *96.6%* |
> > | Adv. Noising (ε=10/255) | 29.9% / *36.9%* | 0.0% / *57.3%* | 0.0% / *11.8%* | 67.0% / *63.3%* |
> > | Adv. Compression        | 2.0% / *1.8%* | 0.8% / *0.5%* | 1.9% / *2.5%* | 6.3% / *5.7%* |
> >
> > The table shows that our Adversarial Compression attack remains effective against all attacks in the presence of double-tail detection. Double-tail detection increases the robustness of DWT and DWT-SVD against adversarial noising, which is the same effect that Jiang et al. found in their paper. Figure 4 of our paper shows that adversarial noising is only effective against Tree-Rings but is ineffective against the remaining methods because an attacker has to add visible noise. An attacker would always use the Adversarial Compression attack in a real-world attack. We will include these results in the revised paper.
> >
> > > Did the authors experiment with different degrees of attacker knowledge -- e.g., what if the attacker does not know the length of the watermark embedded by the actual generator? Would training on 32-bit messages cause attacks on a 64-bit message system to fail?
> >
> > We agree that limiting the attacker’s knowledge can negatively impact the attack’s effectiveness. Limiting knowledge can be done in many ways, for instance, by choosing longer messages, as pointed out by the reviewer. However, this choice can affect trade-offs regarding the watermark’s detectability, robustness, or impact on the quality of the generated images, and choosing such hyperparameters for each watermarking method appropriately is not trivial and outside of the scope of our work. Our attacks operate under the cryptographic assumption that the watermarking method is known (including its parameters, such as the message length) and that only the key-message pair is kept secret. We use the suggested parameters for each watermarking method that the authors are proposing. We will discuss this in the revised paper.
> >
> > > If the attacker "does not need to invoke GKEYGEN" for TRW/WDM/RivaGAN, as stated in section 4.2, does this mean the attacker does not train a differentiable surrogate detector? In this case, doesn't the proposed approach just become a standard white-box attack, as in Jiang et al.?
> > Did the authors train surrogate detectors for all the evaluated watermarking methods to create true no-box attacks?
> >
> > Our attacks always operate without access to the provider’s secret key. Since VERIFY’s response depends on the provider’s secret key, our attacks have no access to the provider’s VERIFY procedure. However, knowledge of the watermarking method allows the attacker to call VERIFY locally (without involving the provider) using surrogate keys generated by the attacker. We also emphasize that our attacks require no access to any of the provider’s generated (and watermarked) images. Our attacker can instantiate their attacks long before a provider deploys their watermarking method.
> >
> > To invoke our attacks, the attacker must only invoke GKEYGEN when VERIFY is not differentiable using keys generated with KEYGEN. If VERIFY is differentiable, they simply use a key generated from KEYGEN. For TRW, WDM, and RivaGAN, the output of KEYGEN are the parameters of a neural network, meaning that VERIFY is already differentiable because we can backpropagate gradients, and there is no need for an additional call to GKEYGEN. Otherwise, as Algorithm 1 shows, the attacker needs to invoke KEYGEN followed by GKEYGEN.
> >
> > We will clarify this in the revised paper.
> >
> > > The authors do not provide any details of the architecture of the surrogate detector networks \theta_d trained by the adversary. This seems like a crucial aspect of the proposed approach, so it is strange that it is not discussed.
> >
> > We always fine-tune a standard ResNet-50 model when invoking GKEYGEN. We will include this in the revised paper.

---

> > > ### Author Response · Authors · 2023-11-13
> > > **additional replies to the comments.**
> > >
> > > > The related work section mixes references to image classifier watermarks [1][2] and watermarks for generative models [3], which are very different: the former aims to protect the intellectual property of a model developer, typically through query- or trigger-based verification, while the latter is embedded in all outputs of a generative model to distinguish real from fake content. This paper is concerned with the latter kind of watermark, so I'm confused by the emphasis on works in the former area. At the very least, these two different types of works should be clearly distinguished from one another in the related work section.
> > >
> > > We categorize watermarking methods by their access to the watermarked medium. White-box watermarking can access the model’s parameters, and black-box methods can query the model on inputs the verifier chooses. As we mentioned in our paper, those have been used primarily for IP protection. In no-box watermarking, the verifier only accesses generated images on attacker-chosen queries, which helps control misuse. Our categorization in Section 2.1 is independent of the task (e.g., image generation or classification) and follows Lukas et al. [C]. We will emphasize this in the paper to avoid any potential confusion.
> > >
> > > > As far as I can tell, the term "Adaptive Attack" comes from the adversarial example literature -- the authors should explain what distinguishes an adaptive attack from a non-adaptive attack and probably cite the original work [4].
> > >
> > > That is an excellent remark, thank you. We will include it in the paper.
> > >
> > > > Figure 1 is missing step #8 (it skips from 7 to 9).
> > >
> > > Thank you for spotting this oversight. We will update the Figure in the revised paper!
> > >
> > > > This is a much smaller concern, but the substitution of the Stable Diffusion v1 generator for v2 does not seem like a very difficult obstacle for the attacker to overcome, given the general similarities in architecture and training. Attacks on post-hoc methods probably shouldn't be affected too much by the choice of surrogate generator, but Tree-Rings is deeply intertwined with the generator structure. Therefore, it would be interesting to see how attacks on TRW fare when there is a more substantial mismatch between the actual and surrogate generator.
> > >
> > > This is an interesting question that we also raise on p.8 - Attack Scalability. We conduct our attack using a best-effort principle by using the least capable public diffusion checkpoint (v1.1) to attack the most capable public diffusion checkpoint (v2.0) at the time. Section 5 details the differences between both checkpoints. We will revise the paper to highlight that our paper only studies diffusion v1.1 and diffusion v2 checkpoints and that the impact this has on any attack’s effectiveness needs to be studied in future work.
> > >
> > > > Rather than training surrogate detectors to reconstruct embedded messages, did the authors consider simply training a binary classifier on watermarked and un-watermarked images from the surrogate generator and then performing an adversarial attack on the binary classifier? This seems like the simplest no-box approach
> > >
> > > This is an interesting point, and we thank the reviewer for raising it. Reconstructing embedded messages includes the ability to detect the presence of a watermark, but the reverse is not necessarily true. The attacker could train simpler detectors by ignoring information about the message while still creating effective attacks. Since such attacks are less capable (and thus cannot be more effective), we do not investigate them.
> > >
> > > > It should be clarified that in the general no-box watermarking scenario, the second step EMBED  need not modify the parameters of the generator model, just endow it with watermarking capabilities (e.g. by applying a post-hoc watermarking algorithm to its outputs). As far as I can tell, none of the methods evaluated in the paper modify the generator parameters directly. Overall, the no-box watermarking steps and their instantiations for each of the evaluated watermarking methods are not clearly explained.
> > >
> > > We agree with the reviewer and will clarify that EMBED can alter the entire generation process, including adding pre- and post-processors.
> > >
> > >
> > > ------
> > >
> > > [C] Lukas, Nils, and Florian Kerschbaum. "PTW: Pivotal Tuning Watermarking for Pre-Trained Image Generators.", 32nd USENIX Security Symposium, (2023).

---

> ### Comment · Reviewer_LPhx · 2023-11-19
> **Reply to authors**
>
> I thank the authors for their detailed reply. I broadly agree with the authors' characterization of the differences between the proposed method and that of Jiang et al. I also appreciate the additional experiments performed against two-tailed detectors. I am willing to increase my score, but I'd like some clarification on points 1-3 below first. Point 4 is less important.
>
> __1.__ I still think the ResNet description is insufficient. For instance, line 6 of Algorithm 1 is doing a lot of heavy lifting. For each of the watermarking methods considered, how is the ResNet configured to extract the encoded message? Does it predict a vector corresponding to the watermark message? Is it trained with cross-entropy or mean squared error loss? This would be good to see in the appendix.
>
> __2.__ Regarding my comment:
> >If the attacker "does not need to invoke GKEYGEN" for TRW/WDM/RivaGAN, as stated in section 4.2, does this mean the attacker does not train a differentiable surrogate detector?...
>
> From the authors' reply, it sounds like a ResNet-based surrogate detector network (or "key") is only trained for DCT and DCT-SVD. For the remaining methods (TRW, WDM, and RivaGAN) the authors presumably train the detector network (or "key") from scratch, as using a provided pre-trained detector network would constitute a white-box attack. So my understanding of the attack is as follows:
> * For DCT/DCT-SVD the attacker must train a ResNet detector from scratch
> * For WDM/RivaGAN the attacker must train a pair of watermark embedder/detector networks from scratch using the corresponding published architectures and training procedures
> * For TRW the attacker uses a pretrained surrogate Stable Diffusion model
>
> Can the authors confirm if this is correct?
>
> __3.__ The actual objective functions for each watermark are not specified in the paper. The VERIFY portion of the objective function will presumably differ from method to method depending on the key and message format -- for example RivaGAN's detector predicts a vector corresponding to the embedded message, while TRW requires measuring the similarity between predicted and known diffusion noise patterns in Fourier space. Again, this would be nice to see in the appendix, as the formulation of an appropriate objective function is a crucial part of any adversarial attack.
>
>
> __4.__ Regarding my comment on the hypothetical surrogate binary classifier attack, I agree that such an attack would operate with less knowledge of the watermark than the proposed method, and should be expected to perform worse. That said, the adversarial attack literature is littered with examples of supposedly naive attacks outperforming more informed baselines due to simplified optimization [1]. I think it would be very interesting to see how such a naive baseline fares, but I don't think such an experiment would be critical to the paper given that the proposed method is already effective.
>
> [1] F. Tramer, N. Carlini, W. Brendel, and A. Madry, "On adaptive attacks to
> adversarial example defenses," in Proc. NIPS, 2020, pp. 1–44.

---

> > ### Author Response · Authors · 2023-11-19
> > **Reply to reviewer**
> >
> > We appreciate the reviewer's response and welcome the opportunity to clarify these questions.
> >
> > > 1. I still think the ResNet description is insufficient. For instance, line 6 of Algorithm 1 is doing a lot of heavy lifting. For each of the watermarking methods considered, how is the ResNet configured to extract the encoded message? Does it predict a vector corresponding to the watermark message? Is it trained with cross-entropy or mean squared error loss? This would be good to see in the appendix.
> >
> > VERIFY consists of a sub-procedure EXTRACT, which maps an image to a message using the secret key, as stated in Section 2.1. The space of messages $\mathcal{M}$ is specific to the watermarking method. All surveyed methods except Tree-Rings are multi-bit watermarks, for which we use the categorical cross-entropy to measure similarity. We configure the ResNet to predict a bit vector of the same length as the message and calculate gradients during training with the cross-entropy loss. As the reviewer suggested, we will add this clarification to the Appendix.
> >
> > > 2. [..] Can the authors confirm if this is correct?
> >
> > Yes, we confirm this is correct and will add this clarification to the Appendix.
> >
> > > 3. The actual objective functions for each watermark are not specified in the paper. [..]
> >
> > We agree and will also add it to the Appendix. There are two objective functions in the paper: one is the categorical cross-entropy for multi-bit messages, and the other is the absolute distance in the Fourier space for Tree-Rings.
> >
> > > 4. I think it would be very interesting to see how such a naive baseline fares, but I don't think such an experiment would be critical to the paper given that the proposed method is already effective.
> >
> > We agree that many modifications could be applied to enhance our method's effectiveness further but leave these to future work since they are outside the scope of the paper. Our focus was to demonstrate that future watermarking methods must consider adaptive attacks if they claim robustness empirically.
> >
> > We will revise the paper shortly and are happy to respond to any further suggestions the reviewer may have. Again, we thank the reviewer for their time and effort to improve our paper.

---

> > > ### Comment · Reviewer_LPhx · 2023-11-19
> > > **Reply to authors**
> > >
> > > I thank the authors for addressing my concerns. I believe the proposed revisions will significantly strengthen the paper, and I have updated my score accordingly.

---

### Official Review · Reviewer_Agid · 2023-10-31

**Soundness:** 2 fair
**Presentation:** 3 good
**Contribution:** 2 fair
**Rating:** 6
**Confidence:** 3

**Summary:**

Authors emphasize the significance of watermarking in countering misuse by marking generated content with hidden messages. The core security property of watermarking, robustness, is investigated in this paper. The authors assert that evaluating robustness involves creating adaptive attacks tailored to specific watermarking algorithms. To this end, one of the paper's contributions is the proposed approach to assess the optimality of adaptive attacks by framing them as an optimization problem and defining an objective function. The paper presents evidence that such attackers can effectively break all five surveyed watermarking methods with negligible degradation in image quality.

**Strengths:**

First, on the aspect of the paper’s organization, this manuscript is well-organized and easy to follow. Second, on the aspect of clarity, the proposed method is clearly defined using schematics and pseudo-code descriptions. Third, this paper provides an approach to evaluating adaptive attacks and the demonstration of their effectiveness provide a fresh perspective on the challenges faced in countering image manipulation.

**Weaknesses:**

The motivation and importance of the proposed method are not clear enough, e.g., what problems did the previous works exist? Besides, the experiments comparison and discussion are weak. Experiment section should expand the scope of discussion, compare with more advanced methods, and provide in-depth discussions.

**Questions:**

1.	ABSTRACT: The text should include more details to the proposed methodology, numerical results achieved, and comparison with other methods
2.	Could you tell me the limitations of the proposed method? How will you solve them? Please add this part to the manuscript.
3.	The abbreviations must appear at the very first place that the terminology is introduced and the way of introducing the terms must be consistent throughout the manuscript from abstract to conclusion.
4.	The conclusions should be improved such that the authors add some analytical terms.

---

> ### Author Response · Authors · 2023-11-13
> **Thank you for your comments!**
>
> We appreciate that the reviewer thinks we provided a “fresh perspective” and for their suggestions to improve our paper further. Please find our detailed response below.
>
> > The motivation and importance of the proposed method are not clear enough, e.g., what problems did the previous works exist?
>
> Watermarking can only effectively control image generator misuse if it is robust. Watermarking methods that claim robustness have been proposed, but we refute their claims and show that their approach to testing robustness is flawed. These methods demonstrate robustness by testing their watermark against attacks that know nothing about the watermarking method, such as JPEG compression, blurring, etc. The issue is that these attacks could perform better if they knew more about the method used to watermark the image, and this leads to a cat-and-mouse game where defenses that claim robustness are proposed, only to be broken later by better (adaptive) attacks.
>
> An attacker who only knows the watermarking method’s algorithmic descriptions, which we refer to as (KEYGEN, EMBED, and VERIFY), can instantiate far more powerful attacks. As our paper shows, instantiating such attacks requires no handcrafting but can be done by leveraging optimization. We define robustness as an objective function, which makes our method generally applicable to any watermarking method proposed in the future. Robustness against adaptive attacks extends to robustness against non-adaptive attacks, which makes studying the former interesting. In cryptography, this principle is known as Kerckhoff’s principle, where an algorithm’s security should not rely on obscurity. Previous works did not consider such adaptive, learnable attackers. We will expand on this discussion in the revised paper.
>
> > Experiment section should expand the scope of discussion, compare with more advanced methods, and provide in-depth discussions.
>
> We will gladly expand the scope of discussion and add comparisons with advanced methods, but we kindly ask for clarification about the methods the reviewer refers to. To our knowledge, there are no strong, comparable attacks without access to the secret watermarking key, as evidenced by watermarking papers [A,B] that only evaluate against attacks such as JPEG compression and quantization. We would greatly appreciate it if the reviewer could point us to “more advanced methods” or detail what part of the discussion we should focus on more in-depth, and we would be happy to revise our paper.
>
> > 1. ABSTRACT: The text should include more details to the proposed methodology, numerical results achieved, and comparison with other methods
> > 4. The conclusions should be improved such that the authors add some analytical terms.
>
> We thank the reviewer for their suggestions and will revise the abstract and conclusion of the revised paper as requested.
>
>  > 2. Could you tell me the limitations of the proposed method? How will you solve them? Please add this part to the manuscript.
>
> We describe our method's limitations to attacking real systems in Section 6. In essence, an adaptive attacker requires access to a (less capable) surrogate generator and the algorithmic description of the watermarking method (meaning KEYGEN, EMBED, and VERIFY). As stated in the paper, we took a best-effort approach. We used the least capable public diffusion model (Stable Diffusion v1.1) to remove watermarks from the most capable public diffusion model (Stable Diffuson v2.0). We also discuss in the paper that the watermarking algorithm may only sometimes be released. Still, the provider relies on security through obscurity and remains vulnerable to attacks by anyone to whom this information is released.
>
> Furthermore, Section 6 also describes limitations on the attacks that we consider and that more powerful attacks may exist that could be considered in future work.
>
> > 3. The abbreviations must appear at the very first place that the terminology is introduced and the way of introducing the terms must be consistent throughout the manuscript from abstract to conclusion.
>
> We apologize for any confusion if an abbreviation was used before defining it. We are happy to revise and clarify the paper if the reviewer could point us to the specific abbreviations that have caused confusion.
>
> We are happy to resolve any other questions the reviewer may have. If these are the only concerns, we kindly ask the reviewer to consider increasing their score.
>
> ---------
>
> [A] Wen, Yuxin, et al. "Tree-Ring Watermarks: Fingerprints for Diffusion Images that are Invisible and Robust." arXiv preprint arXiv:2305.20030 (2023).
>
> [B] Zhao, Yunqing, et al. "A recipe for watermarking diffusion models." arXiv preprint arXiv:2303.10137 (2023).

---

> > ### Author Response · Authors · 2023-11-22
> >
> > Dear Reviewer Agid,
> >
> > We appreciate all of the valuable time and effort you have spent reviewing our paper. As today is the last day of the discussion period, we kindly ask that you review our reply and consider updating your score if needed. We believe that we have addressed all questions and concerns raised, but please feel free to ask any clarifying questions you might have before the end of the discussion period.
> >
> > Best,
> > The Authors

---

### Official Review · Reviewer_f7pQ · 2023-10-31

**Soundness:** 3 good
**Presentation:** 3 good
**Contribution:** 3 good
**Rating:** 6
**Confidence:** 5

**Summary:**

Untrustworthy users can misuse image generators to synthesize high-quality deep-
fakes and engage in online spam or disinformation campaigns. Watermarking de-
ters misuse by marking generated content with a hidden message, enabling its
detection using a secret watermarking key. A core security property of water-
marking is robustness, which states that an attacker can only evade detection by
substantially degrading image quality. Assessing robustness requires designing
an adaptive attack for the specific watermarking algorithm. A challenge when
evaluating watermarking algorithms and their (adaptive) attacks is to determine
whether an adaptive attack is optimal, i.e., it is the best possible attack. We solve
this problem by defining an objective function and then approach adaptive attacks
as an optimization problem. The core idea of our adaptive attacks is to replicate
secret watermarking keys locally by creating surrogate keys that are differentiable
and can be used to optimize the attack’s parameters. The authors demonstrate for Stable
Diffusion models that such an attacker can break all five surveyed watermarking
methods at negligible degradation in image quality. These findings emphasize the
need for more rigorous robustness testing against adaptive, learnable attackers.

**Strengths:**

Paper is well formatted

Topic is interesting

Good balance of theory and experiments

**Weaknesses:**

Please improve readability

Please number all equations

Please discuss figures, tables and algorithms clearly in the text

Please add a security analysis to known attacks in this domain

**Questions:**

Why is this topic important?

What are the future work directions of this work?

Why is the comparative analysis limited

What is the complexity of the algorithms?

---

> ### Author Response · Authors · 2023-11-13
> **Thank you for your comment!**
>
> We thank the reviewer for their positive assessment of our work and comments which will help further improve our paper. Please find answers to all questions below.
>
> > Please improve readability. Please number all equations. Please discuss figures, tables and algorithms clearly in the text.
>
> Thank you for bringing to our attention that some equations are not labeled. We will revise the paper to label all equations and ensure that all Figures, Tables, and Algorithms are discussed in the text.
>
> > Why is this topic important?
>
> As stated in the introduction, high-quality image generators can be misused by untrustworthy users, leading to the proliferation of deepfakes and online spam. This is already a problem on social media [A] and has led to companies such as Google announcing their image watermark, such as SynthID [B]. Also, the US government has released an “AI executive order” [C] asking providers to watermark generated outputs. Watermarking needs to be robust; otherwise, attackers can easily evade it by removing the watermark from an image with imperceptible modifications. We propose a better method to test the robustness of watermarking methods using adaptive attacks that are designed against a specific watermarking method. We will include the above references in the revised paper.
>
> > What are the future work directions of this work?
>
> The future direction of our work is to use our adaptive attacks to design watermarking methods that withstand them. Such a watermarking method’s claim to robustness is more convincing, as robustness to adaptive attacks extends to robustness against non-adaptive attacks. We will include this point in the revised paper’s discussion section.
>
> > Why is the comparative analysis limited?
>
> We believe that we have mentioned all related work in this field. Our work is the first to instantiate _adaptive_ attacks against image watermarking where the attacker knows the watermarking method but not the secret key or the message. Our attacks require no access to the provider’s watermark verification procedure (unlike other works, such as Jiang et al.). The proposed attacks can remove a watermark from a single image, regardless of which key-message pair was used to watermark it. We would appreciate it if the reviewer would further specify in what regards they believe our comparative analysis is limited. We will be happy to address it in the revised paper.
>
> > What is the complexity of the algorithms?
>
> Thank you for this question. From a computational perspective, generating a surrogate watermarking key with methods such as RivaGAN or WDM is the most expensive operation, as it requires training a watermark encoder-decoder pair from scratch. Generating a key for these two methods takes around 4 GPU *hours* each on a single A100 GPU, which is still negligible considering the total training time of the diffusion model, which takes approximately 150-1000 GPU *days* [D]. The optimization of Adversarial noising takes less than 1 second per sample, and tuning the adversarial compressor’s parameters takes less than 10 minutes on a single GPU. We will revise the paper to include the measured running times in the Appendix of the revised paper.
>
> We hope that we have addressed all of the reviewer’s questions and would kindly ask the reviewer to increase their score if these are the only questions.
>
> ------
>
> [A] Barrett, Clark, et al. "Identifying and mitigating the security risks of generative ai." arXiv preprint arXiv:2308.14840 (2023).
>
> [B] Sven Gowal and Pushmeet Kohli. Identifying ai-generated images with SynthID, 2023. URL https://www.deepmind.com/blog/identifying-ai-generated-images-with-synthid. Accessed: 2023-09-23.
>
> [C] "Safe, Secure, and Trustworthy Development and Use of Artificial Intelligence." Federal Register, 2023, https://www.federalregister.gov/documents/2023/11/01/2023-24283/safe-secure-and-trustworthy-development-and-use-of-artificial-intelligence. Accessed 4 Nov. 2023.
>
> [D] Dhariwal, Prafulla, and Alexander Nichol. "Diffusion models beat gans on image synthesis." Advances in neural information processing systems 34 (2021): 8780-8794.

---

> > ### Comment · Reviewer_f7pQ · 2023-11-22
> >
> > Thanks to the authors for addressing my comments

---

### Author Response · Authors · 2023-11-17
**Summary of revisions**

We once again express our sincere gratitude for the time that all reviewers spent into providing feedback to improve our paper. We have incorporated these valuable suggestions into our revised paper, as promised in our rebuttal, which we believe has further strengthened our work.

At your convenience, we have highlighted all modifications in blue. We remain open to further discussions or additional revisions as suggested by the reviewers. Below, please find a summary of the changes we applied for the revision.

--------

## Main paper
**Abstract & Introduction**
* (_R2_) Added the requested details to the abstract (and conclusion)

**Section 2 - Background**
* (_R3_) Added a remark that our categorization is independent from the task
* (_R3_) Clarified the definition of a key and message

**Section 6 - Discussion**
* (_R1_, _R2_)  Emphasized the importance of studying adaptive attacks (including the anticipated impact on future work)
* (_R3_)  Highlight differences between our work and that of Jiang et al.
* (_R2_, _R3_) Emphasize limitations (e.g., that we only study diffusion v1.1 and v2 models)
* (_R2_, _R4_) Added motivation for studying adaptive attacks. These remain relevant as a tool for the provider to test the robustness of their watermark even when the watermarking method will not be released
* (_R3_) Highlight that our work studies the best public watermarking parameters suggested by the authors.
* (_R4_) Added that our attacks should not break the robustness of the surveyed watermarks if KEYGEN was sufficiently randomized, even if the attacker has access to the same model as the provider

## Minor

* (_R1_) Numbered all equations and ensured that Figures, Tables and Algorithms are discussed in the text.
* (_R3_) Fixed an oversight in Figure 1.
* (_R3_) Clarified that EMBED can alter the entire generation process, including adding pre- and post-processors.
* (_R3_) Added a reference to Carlini et al. for adaptive attacks.
* (_R3_) Add a remark that we always fine-tune ResNet-50 when invoking GKEYGEN

## Appendix

* (_R1_) Included an "Attack Efficiency" Section describing the running time of our attacks.
* (_R3_) Included the results using double-tail detection.
* (_R3_) Included details on Algorithm 1 (GKEYGEN)

Due to the page limit, we moved Table 2 into the Appendix.

---

### Meta-Review · Area_Chair_xfhi · 2023-12-07

**Metareview:**

The paper presents a novel approach to evaluating the robustness of different watermarking methods for generative models, with a focus on adaptive attacks using differentiable surrogates. It delves into both theoretical aspects and practical applications, making it a timely contribution to the field. Reviewers appreciated the paper's clear organization, straightforward presentation of methods, and the inclusion of interesting experiments. However, they also pointed out several weaknesses, such as the need for clearer motivation, more detailed discussion of experiments, and a broader comparison with advanced methods. The terminology used in the paper was another point of concern, as it could lead to confusion among readers. The authors have made considerable efforts to address these issues in their revised submission, as evidenced by the their responses.

**Justification For Why Not Higher Score:**

A higher score for this paper is not justified primarily because of the limited scope of the experiments. While the revisions have addressed some of these concerns, there are still gaps in the comparison with more advanced methods and a comprehensive discussion of the experimental results.

**Justification For Why Not Lower Score:**

Despite its shortcomings, the paper should not receive a lower score as it addresses a significant issue in the field of generative models — the robustness of watermarking methods. The novel approach of using differentiable surrogates for adaptive attacks is both interesting and valuable.

---

### Decision · Program_Chairs · 2024-01-16

Accept (poster)